# A Floating Gate Memory with U-Shape Recessed Channel for Neuromorphic Computing and MCU Applications

**DOI:** 10.3390/mi10090558

**Published:** 2019-08-23

**Authors:** Lu-Rong Gan, Ya-Rong Wang, Lin Chen, Hao Zhu, Qing-Qing Sun

**Affiliations:** State Key Lab. of ASIC and System, School of Microelectronics, Fudan University, Shanghai 200433, China

**Keywords:** U-shape recessed channel, floating gate, neuromorphic computing, MCU (microprogrammed control unit)

## Abstract

We have simulated a U-shape recessed channel floating gate memory by Sentaurus TCAD tools. Since the floating gate (FG) is vertically placed between source (S) and drain (D), and control gate (CG) and HfO_2_ high-k dielectric extend above source and drain, the integrated density can be well improved, while the erasing and programming speed of the device are respectively decreased to 75 ns and 50 ns. In addition, comprehensive synaptic abilities including long-term potentiation (LTP) and long-term depression (LTD) are demonstrated in our U-shape recessed channel FG memory, highly resembling the biological synapses. These simulation results show that our device has the potential to be well used as embedded memory in neuromorphic computing and MCU (Micro Controller Unit) applications.

## 1. Introduction

With the popularity of intellectualization in medical devices, automotive electronics, smart grid, green energy, wearing equipment, smartcards, and the rise of the Internet of things, Microprogrammed Control Unit (MCU) has been widely used in industrial control and consumer electronics markets and has shown tremendous growth potential in the next few years. To reduce peripheral discrete devices and increase applicability, MCU tends to store programs and small amounts of data through embedded non-volatile memory (NVM). Therefore, with the expanding scale of semiconductor devices and the increasing density of transistors, embedded flash memory, as an important branch of flash products, is more and more widely used in the booming MCU market, and its requirement of integration density is higher and higher [1,2,3,4]. With the development of Moore’s law, the traditional horizontal channel embedded flash memory has limited miniaturization capability and encountered the small size effect. The leakage caused by this effect will affect the memory’s judgment of 0/1 state, which is a serious problem to be avoided, especially in the development of multi-value storage of floating gate (FG) memory.

Today digital computers are based on von Neumann architecture where the memory and processor are physically separated. This fundamentally limits the development of modern computers [5]. Envisioned by Carver Mead in 1990, neuromorphic computing seeks inspirations from the massive parallelism, robust computation, and high energy efficiency of the human brain and can potentially give rise to a revolutionary computing technology that fundamentally overcomes the von Neumann bottleneck in conventional digital computers [6,7,8,9,10]. Synapse is the basic unit in biological nervous system, which connects between two neurons and response differently to incident signals [11]. The change of the strength of synaptic weights caused by memorization events is in charge of encoding and storing memory. Mimicking the physiological synaptic behaviors by using electronic devices is the most important step for neuromorphic systems [12]. The embedded flash memory can emulate the synaptic behaviors such as long-term potentiation (LTP) and long-term depression (LTD), and a high accuracy of more than 1% can be obtained in the application of neuromorphic computing [13]. However, the slow operation speed of traditional embedded floating-gate memory and its limited miniaturization ability hinder its further development in neuromorphic computing [14].

For the first time, this paper proposes a new FG memory structure (UFGM) based on NAND flash programming method and U-shape recessed channel for the applications of neuromorphic computing and MCU. Since the floating gate is vertically placed between source and drain, and control gate and HfO_2_ high-k dielectric extend above source and drain, the integrated density can be well improved. The enlarged tunneling area and enhanced tunneling rate dramatically increase the tunneling current when the device is turned on, and the erasing and programming speed of the device are respectively decreased to 75 ns and 50 ns. Therefore, UFGM can quickly adjust synaptic weights during long-term potentiation (LTP) and long-term depression (LTD) operation. In addition, the off-leakage current of UFGM is suppressed because of the extended physical channel length [15,16,17,18], which is conducive to reducing the power consumption whether it is used as a synaptic device in the application of neuromorphic computing or MCU. Furthermore, for UFGM, because FG is U-shape embedded, there is no FG capacitive coupling crosstalk between cell and cell in the storage matrix.

## 2. Device Structure 

We have simulated two devices with Sentaurus TCAD tools. Their difference is the doping type of FG. The first device structure is shown in Figure 1a and its FG is $p^{+}$-doped. The second device structure is shown in Figure 1b and its FG is $n^{+}$-doped.

Take the first device as an example. The $p^{+}$-doped FG is buried vertically between source (S) and drain (D), and S and D are cut off, and the channel becomes U-shape recessed. This can save area to increase device density, reduce short-channel effects, reduce cell-to-cell coupling, and suppress the off-leakage current. These features will facilitate the applications of UFGM in neuromorphic computing and MCU.

The traditional SiO_2_ blocking layer between the polysilicon control gate (CG) and $p^{+}$-doped FG is replaced with 12 nm HfO_2_ high-k dielectric, and CG and HfO_2_ high-k dielectric extend above S and D. The advantage of this is that the inversion and accumulation of electrons and holes on both sides of S and D can be directly controlled by CG through HfO_2_ high-k dielectric, which will greatly enhance Fowler–Nordheim (F-N) tunneling rate. Another advantage is that FG is coupled to CG directly through HfO_2_ high-k dielectric, and the coupling capacitance is increased, so the CG potential can be dropped to FG more effectively, thus enhancing FN tunneling rate. In terms of the tunneling area, UFGM also shows its advantage. Compared with the horizontal channel, the U-shape recessed channel can increase the effective tunneling area approximately twice under the same feature size. The enlarged tunneling area and enhanced tunneling rate can dramatically increase the tunneling current when the device is turned on. 

We also simulated two devices with original SiO_2_ based FG for comparison. The first device structure is shown in Figure 2a and its FG is $p^{+}$-doped. The second device structure is shown in Figure 2b and its FG is $n^{+}$-doped. The fabrication process of the device is similar to that of the UFGM with HfO_2_ based FG, except that the 12 nm HfO_2_ high-k dielectric material is replaced by 12 nm SiO_2_.

## 3. Electrical Characteristics

Table 1 contains the main physical models used in electrical simulation. The non-local tunneling model is powerful. It can deal with any shape of barrier and take into account the carrier heating. It allows users to describe tunneling between valence band and conduction band, and approximates several different tunneling probabilities. Non-local tunneling includes FN tunneling.

We studied the change in the FG potential during one operation period. There are similar trends in the two kinds of devices. As described in Figure 3, under the same conditions, the amount of change in the FG potential gradually increases as $V_{cg}$ increases. Due to the capacitive coupling, a change in the FG potential will cause a drift in the device threshold voltage, which will be used to distinguish between state 0 and state 1 during the reading operation. In the erasing/programming operation, there is a balance between the voltage magnitude and the time setting. Take the UFGM with $p^{+}$ FG as an example, at $V_{CG}$ = 10 V and time = 50 ns, the FG potential drops by 0.0528 V, while at $V_{CG}$ = 13 V and time = 50 ns, the FG potential drops by 1.8527 V. However, by extending the bias time at $V_{CG}$ = 10 V, we can get the same FG potential change as at $V_{CG}$ = 13 V and time = 50 ns. The erasing and writing speed can be manually adjusted with different voltage and the time of reading and writing sequence. Therefore, the specific setting of working voltage and time should be carried out under the specific requirements of high speed or low power design.

There are also some slight gaps between two kinds of devices. In the programming operation, the amount of change in the FG potential of the UFGM with $p^{+}$ FG is much larger than the UFGM with $n^{+}$ FG, which means the UFGM with $p^{+}$ FG responds faster. For example, at $V_{CG}$ = 15 V and time = 50 ns, the FG potential of the UFGM with $p^{+}$ FG drops by 4.6726 V and the FG potential of the UFGM with $n^{+}$ FG drops by 4.4548 V. In the erasing operation, the amount of change in the FG potential of the UFGM with $n^{+}$ FG is much larger than the UFGM with $p^{+}$ FG, which means the UFGM with $n^{+}$ FG responds faster. For example, at $V_{CG}$ = −15 V and time = 50 ns, the FG potential of the UFGM with $p^{+}$ FG increases by 0.1327 V and the FG potential of the UFGM with $n^{+}$ FG increases by 0.2259 V. As a conclusion, these two devices have their own advantages. In the application of neuromorphic computing and MCU, we can choose the suitable device according to actual needs.

We also studied the change of FG potential of UFGM based on SiO_2_ under different operating voltage. There is a similar trend in these two devices. As shown in Figure 4a, the variation of FG potential increases with the increase of $V_{cg}$ in programming operation. At $V_{cg}$ = 15 V, the potential change of $p^{+}$ UFGM based on SiO_2_ is 0.2637 V, but at the same voltage, the potential change of $p^{+}$ UFGM based on HfO_2_ can reach 4.6726 V. When $V_{cg}$ = 20 V, the potential change of $p^{+}$. UFGM based on SiO_2_ can reach 4.1173 V, which is still lower than that of UFGM based on $p^{+}$ HfO_2_ when $V_{cg}$ = 15 V. As shown in Figure 4b, in the erasing operation, the change in FG potential gradually decreases as $V_{cg}$ increases. At $V_{cg}$ = −20 V, the potential change of $n^{+}$ UFGM based on SiO_2_ is 3.2647 V while the potential change of $n^{+}$ UFGM based on HfO_2_ can reach 3.6059 V at the same operation voltage. By comparing the potential changes, we can find that UFGM based on HfO_2_ has obvious speed advantages over UFGM based on SiO_2_ in both programming and erasing operations.

Figure 5a,b describes the change of the FG potential with time, and the operating voltage scheme as shown in Table 2. During the programming operation, as described in Figure 5a, potential gradually decreases as time increases. The potential decreases approximately linearly in the first 1 μs, and with the increase of time, the potential decreases slowly and finally tends to saturation state. However, the time of linear change of potential is close to 1 μs, and the change of FG potential is about 2.0212 V, which is already enough to distinguish state 0 and state 1. For example, in this paper, we only need 50 ns of operation time. In LTP/LTD operations, there is also sufficient time for weights to approximate linear variations. Similarly, during the erasing operation, as described in Figure 5b, FG potential gradually increases as time increases and the potential increases approximately linearly in the first 1 μs. With the increase of time, the potential increases slowly and finally tends to saturation state. The time of linear change of potential during erasing operation is close to 1.6 μs, and the change of FG potential is about 1.4957 V, which is also enough to distinguish 0/1 state.

Figure 6 is the drain current ($I_{d}$) curve of UFGM cell extracted in the second cycle. The operation voltage and time settings of UFGM with $p^{+}$ FG are given in Table 2. According to the simulation experience, the current is more stable and reproducible from the second cycle. The drain current curve of UFGM with $p^{+}$ FG and UFGM with $n^{+}$ FG cells are shown in Figure 6a,b, respectively. There are also similar trends in the two kinds of devices. As can be seen from Figure 6a, after 50 ns programming operation, a small $I_{d}$ of about 1.84 × 10^−8^ A can be read and state 0 is successfully written. After 75 ns erasing operation, a large current of about 1.42 × 10^−6^ A can be read under the same reading voltage, and state 1 is successfully written. The $I_{ON}/I_{OFF}$ ratio is over 77. As can be seen from Figure 6b, after 50 ns programming operation, a small $I_{d}$ of about 1.01 × 10^−9^ A can be read and state 0 is successfully written. After 75 ns erasing operation, a large current of about 3.81 × 10^−7^ A can be read under the same reading voltage, and state 1 is successfully written. The $I_{ON}/I_{OFF}$ ratio is over 376. In the application of MCU, we need to distinguish the state “0” and the state “1” as clearly as possible, so the difference value between $I_{ON}$ and $I_{OFF}$ should be as large as possible to achieve this distinction, so it is more appropriate to use the UFGM with $p^{+}$ FG at this time. In the application of neuromorphic computing, for example, we build a neural network to do weight updates, the $I_{ON}/I_{OFF}$ ratio should be as large as possible to get as many adjustable current states as possible. Here, the UFGM with $n^{+}$ FG is more suitable. From the simulation results, we can see that a high-speed embedded FG memory with good characteristics of scaling down is realized, which has the potential to be well applied to neuromorphic computing and MCU.

In the biological brain, the energy efficiency of synaptic transmission is not fixed, which changes with the change of synaptic activity pattern. In many synapses, repeated stimuli can produce an increase or decrease in synaptic weights up to hours or even days. Synaptic weights refer to the strength or magnitude of synaptic weights between the presynaptic and postsynaptic nodes. The enhancement of synaptic weight is called long-term potentiation (LTP), and the reduction of synaptic weight is called long-term depression (LTD). LTP and LTD are the material basis for learning and memory formation [19]. We use the UFGM with $p^{+}$ FG as an example to simulate the LTP and LTD characteristics of synapses.

Figure 7 shows the LTP and LTD characteristics of UFGM with $p^{+}$ FG. The operation voltage and time settings of pulses applied to UFGM with $p^{+}$ FG are given in Table 2. Each erasing/programming pulse is followed by a 1 ns read pulse to monitor the erasing/programming effect. As shown in Figure 7a, the current flowing through the device increases with the increase of the number of pulses, which means the UFGM with $p^{+}$ FG exhibits obvious LTP characteristics under a series of pulses with a width of 1.5 ns width and an amplitude of −15 V. Changing the direction of the programmable pulse, setting the pulse width to 1 ns, the amplitude to 11 V, the current flowing through the device decreases gradually with the increase of the number of pulses, and the device shows obvious LTD characteristics. As shown in Figure 7b, when the pulse width is 1.5 ns, the amplitude is −15 V, the potential of the FG increases with the increase of the number of pulses, and the threshold voltage of the device reduces gradually. At a constant reading voltage, the device shows obvious LTP characteristics. Similarly, by changing the direction of the programmed pulse, the device is stimulated by a pulse with a pulse width of 1 ns, an amplitude of 11 V. The potential of the device decreases gradually with the increase of the number of pulses, thus the threshold voltage of the device increases gradually. At the same constant reading voltage, the distinct LTD characteristics can be displayed. 

The linearity in weight update refers to the linearity of the curve between the device conductance and the number of identical programming pulses. Ideally, this should be a linear and symmetrical relationship that maps the weight of the algorithm directly to the conductance of the device [14]. This nonlinearity/asymmetry is undesirable because the weight changes depend on the current weight, or in other words, the weight updates are historically relevant [20,21,22]. As can be seen from Figure 7, the drain current and potential curves of UFGM with $p^{+}$ FG have good linearity and symmetry, which means the weight update of UFGM with $p^{+}$ FG has excellent linearity and symmetry. This can avoid the loss of learning accuracy of neural networks due to nonlinearity/asymmetry.

## 4. Conclusions

In this research, we designed and simulated two new structures of U-shape recessed channel FG memory using Sentaurus TCAD tools. After 50 ns programming operation and 75 ns erasing operation, the $I_{ON}/I_{OFF}$ ratio of the UFGM with $p^{+}$ FG is over 77, while the $I_{ON}/I_{OFF}$ ratio of the UFGM with $n^{+}$ FG is over 376. When a series of continuous pulse operations are applied, the UFGM shows obvious LTP and LTD characteristics. The increase in operating speed, the decrease in short-channel effects and cell-to-cell coupling of FG, the enhanced tunneling rate, the excellent LTP and LTD characteristics, and the increased scaling down ability of the device due to structural changes, make it suitable for the use as an embedded FG memory in neuromorphic computing and MCU.

## Figures and Tables

**Figure 1 micromachines-10-00558-f001:**
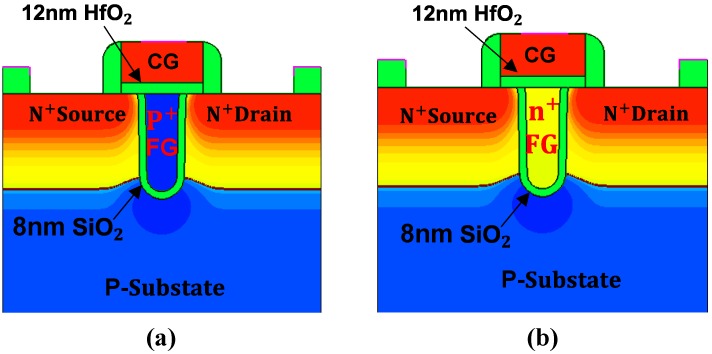
The device structure of (**a**) a new FG memory structure UFGM with $p^{+}$ floating gate (FG) and (**b**) UFGM with $n^{+}$ FG.

**Figure 2 micromachines-10-00558-f002:**
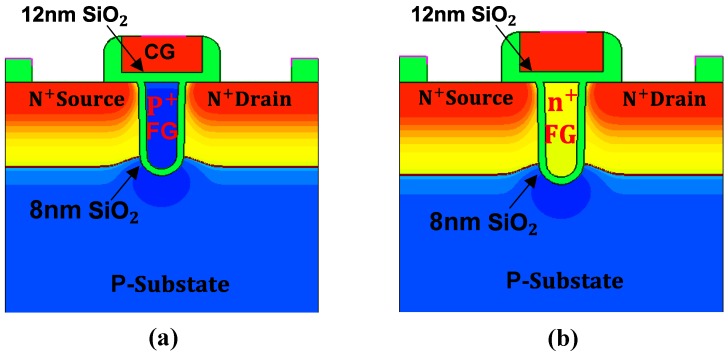
The device structure of UFGM with SiO_2_ based (**a**) $p^{+}$ FG and (**b**) $n^{+}$ FG.

**Figure 3 micromachines-10-00558-f003:**
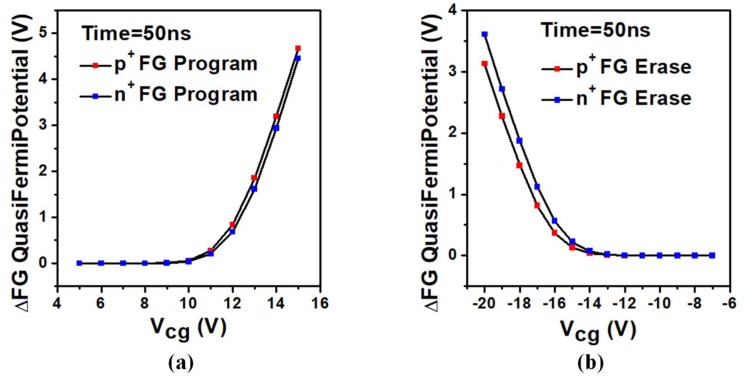
FG potential shift in UFGM as a function of $V_{CG}$ after (**a**) 50 ns programming operation and (**b**) 50 ns erasing operation. The other contacts are set to 0 V.

**Figure 4 micromachines-10-00558-f004:**
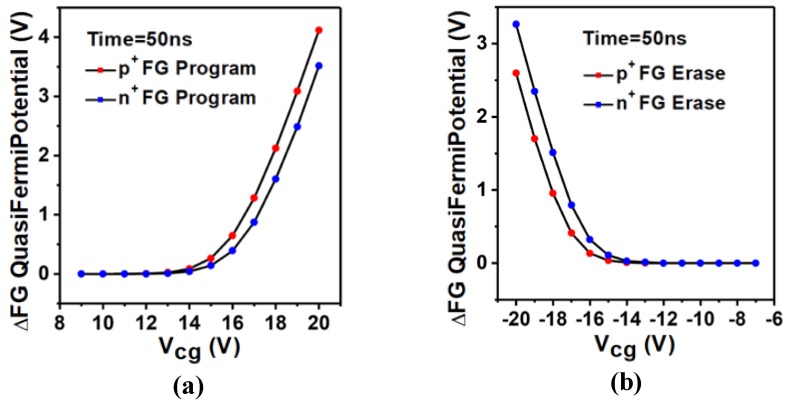
FG potential shift in UFGM with SiO_2_ based FG as a function of $V_{CG}$ after (**a**) 50 ns programming operation and (**b**) 50 ns erasing operation. The other contacts are set to 0 V.

**Figure 5 micromachines-10-00558-f005:**
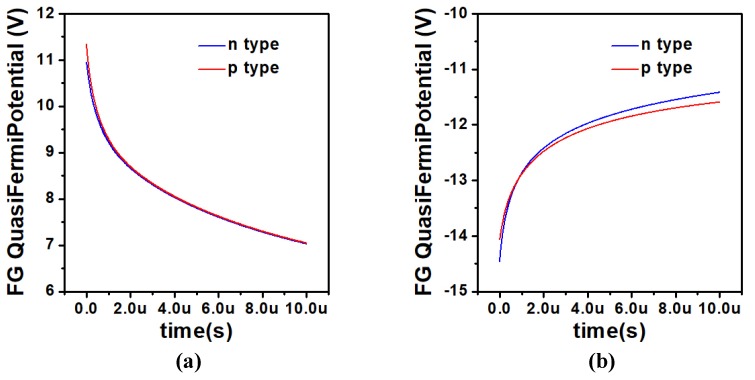
FG potential in UFGM as a function of time after (**a**) programming operation and (**b**) erasing operation using the operation voltage scheme in Table 2

**Figure 6 micromachines-10-00558-f006:**
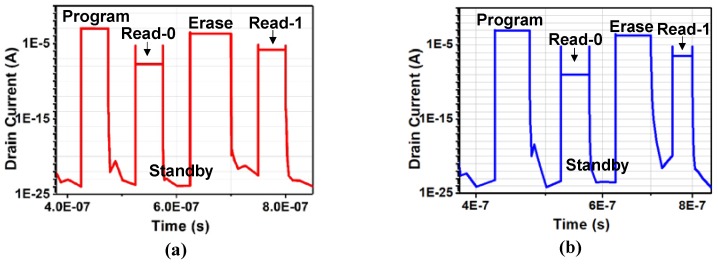
The $I_{d}$ change curve of (**a**) UFGM with $p^{+}$ FG and (**b**) UFGM with $n^{+}$ FG with time in a transient simulation using the operation voltage scheme in Table 2.

**Figure 7 micromachines-10-00558-f007:**
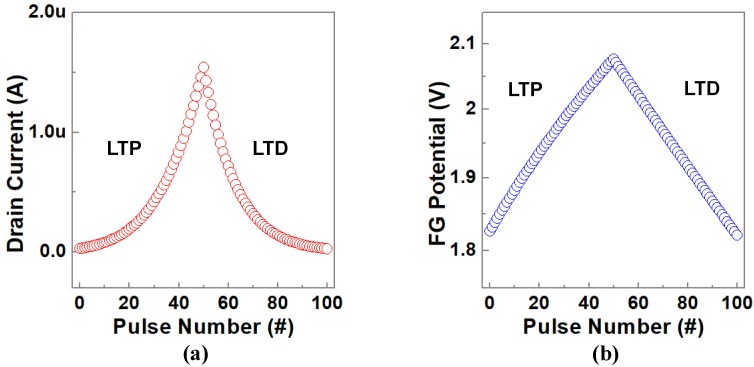
Long-term potentiation (LTP)/long-term depression (LTD) characteristics of UFGM with $p^{+}$ FG: (**a**) Drain current and (**b**) FG potential vary with the number of pulses in a transient simulation using the operation voltage scheme in Table 2.

**Table 1 micromachines-10-00558-t001:** Main physical models selection.

Interface	Physical Mechanism	Model Selection
Oxide/FG poly	Nonlocal tunneling	eBarrierTunneling hBarrierTunneling
Oxide/silicon	Nonlocal tunneling	eBarrierTunneling hBarrierTunneling

**Table 2 micromachines-10-00558-t002:** Operation voltage and time of UFGM with $p^{+}$ FG.

Voltage or Time	Program	Erase	Read	Standby
*V_CG_* (V)	11	−15	1.5	0
*V_D_* (V)	0	0	2	0
*V_S_* (V)	0	0	0	0
*V_Sub_* (V)	0	0	0	0
Figure 6 Time (ns)	50	75	50	50
Figure 7 Time (ns)	1	1.5	1	2

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
