# Peer review of "A Floating Gate Memory with U-Shape Recessed Channel for Neuromorphic Computing and MCU Applications"

_micromachines, 2019, doi:10.3390/mi10090558_

Round 1
Reviewer 1 Report
- Authors described a recessed channel strcutrue as U-shaped floating gate. For more versatile point of definition, authors are required to call the device as recessed channel device. At least, author should call it simultatneously. Is there any critical point for the synaptic function with the round corner of U-shape? If not, it is just another kind of recessed channel.
- For an availability for the synaptic operation, the key parameter should be physical properties of P+ FG and N+ FG. Authors should elaborate to explain the electrical properties which authors uased in simulations. Without these discussions, no meaningful paper can be accomplished.
- Since long term potentiation could be shown in simulation, short term potentiation(STP) can be also represented. Authors are required to show the STP without many difficulties in terms of pulse time width or delay time variations. Even it is a simulation. Having both LTP and STP should be a good combination to highlight the results.
Reviewer 2 Report
This paper proposed a proposes a floating gate memory structure with U-shaped channel for next generation computing. This work combines U-channel and HfO2 oxide layer to propose a new FG. Though the work seems interesting, but everything presented in this paper is based on simulation-based analysis. Therefore, actual fabrication of this device is necessary to validate authors claim. In addition, the organization of the paper can be improved.
Big items:
-This paper is purely based on the simulation-based analysis. In my view simulation-based analysis is not sufficient to determine the actual characteristic of a floating gate memory (FGM) structure. Especially, this work did not discuss fabrication feasibility of this gate at all. For example, is it possible to deposit HfO2 along with other materials in the construction of FGM. Therefore, fabricating a sample FGM is necessary to analyze actual working of this gate.
-The discussion on the related works in introduction section can be improved. Especially the authors need to clearly motivate why there is a need for proposed FGM.
-It is better to compare various parameters presented in Table 1 of proposed FGM with original SiO2 based FG such that reader appreciate the contributions of the authors.
Small Things:
-Page 1 of 6: MCU, LTP, and LTD needs to be defined on their first sighting in the introduction Section. Please define them.
-Figures 1-4: Sub-figure indentation like (a), (b) are placed on the top left corner of the figure, I believe they need to be in bottom center of each figure.
-Section 3: Make sure section title and its starting text in the same page
Round 2
Reviewer 1 Report
- Most of issues commented on 1st review were resolved.
Reviewer 2 Report
Authors did a good job in addressing most of my comments. It is recommended to address following concern before considering this paper for publication.
I still believe that fabrication is necessary to determine the actual characteristic of a floating gate memory (FGM) structure. Atleast in this work authors need to add a discussion about the feasibility of HfO2 deposition along with other materials in the construction of FGM.